# Moderating Effects of Internationalization between Corporate Social Responsibility and Financial Performance: The Case of Construction Firms

**Meiyue Sang** [1] , **Yuqing Zhang** [1] **, Kunhui Ye** [1] **and Weiyan Jiang** [2,*]

[1] School of Management Science and Real Estate, Chongqing University, Chongqing 400045, China; MeiyueSang@cqu.edu.cn (M.S.); YuqingZhang@cqu.edu.cn (Y.Z.); Kunhui_YE@Cqu.edu.cn (K.Y.)
[2] Business School, Southwest University of Political Science and Law, Chongqing 401120, China
* Correspondence: jiangweiyan@swupl.edu.cn

**Abstract:** The relationship between corporate social responsibility (CSR) and corporate financial performance (CFP) has been crucial in academia and business circles. Numerous construction firms have continued to internationalize construction business over time despite the influence of the COVID-19. The internationalization of construction business makes the CSR–CFP relationship more complicated than usual. Construction firms' CSR fulfillment serves to engage in reliable relationships with stakeholders and consequently improve CFP. It can bring both benefits and costs to the firm, which suggests that the CSR–CFP relationship is non-linear. This study examines the impacts of CSR on the financial performance of construction firms. We took Chinese-listed construction companies as an example, and an inverted U-shaped curve relationship between CSR and CFP was eventually revealed. Further, the significant moderating role of the degree of corporate internationalization (DOI) in the CSR–CFP relationship is disclosed. The results show that matching a high DOI-high CSR and a low DOI-low CSR is more conducive to CFP promotion. Thus, this research makes contributions to the academic perception of the impacts of CSR and DOI on CFP and provides insights for CSR fulfillment in the international arena.

**Keywords:** corporate social responsibility (CSR); corporate financial performance (CFP); internationalization; construction business

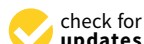



## 1. Introduction

The importance of corporate social responsibility (CSR) and its performance evaluation has been highlighted by both business and academic professionals over time [1]. Although researchers have maintained enthusiasm regarding the relationship between CSR and corporate financial performance (CFP), the results are inconclusive and contradict each other. Some researchers have pointed out that CSR–CFP embodies a positive linear correlation [2,3]. The fulfillment of CSR helps firms to strengthen their relationships with stakeholders and acquire resources to improve their financial performance [4]. Therefore, several studies have emphasized that undertaking CSR activities to improve firm reputation, reduce business operation risks, and fortify innovation capabilities can be valuable [5]. Alternatively, CSR activities cause resource consumption and agency costs may have a negative impact on CFP [6,7]. In this case, recent studies have found a U-shaped relationship [8,9] or inverted U-shaped relationship [10,11] between CSR and CFP, which indicates that they may have a complex relationship.

The construction industry contributes significantly to sustainable development by creating job opportunities and promoting economic growth [12]. However, the overall impact of construction production is substantially negative due to the generation of waste and pollution, damage to the ecosystem, and safety and occupational health problems for people [13]. The emergence of economic globalization has resulted in construction

firms pursuing "going out" strategies, and "international construction" has become a business operation paradigm [14]. CSR is a firm competitiveness factor [15]. Currently, multinational firms have to respond to the needs of stakeholders and assume greater social responsibilities based on legitimacy and strategic motives [16]. CSR engagement favors firms to alleviate the conflicts of cultural, political, economic, and social differences caused by internationalization [17]. Firms that have newly entered the international market have lower comparative popularity, and they might face "legitimacy" assessments based on limited information, prejudices, or stereotypes [18,19]. At this moment, implementing CSR facilitates firms to change the disadvantaged situation. It is not a surprise that firms having extensive international businesses presence may conduct greater CSR efforts, mobilize stakeholders' support [20], and continue to gain legality in overseas markets. Therefore, the degree of internationalization (DOI) is assumed to be crucial in moderating the CSR–CFP relationship.

According to stakeholder and agency theories, fulfilling CSR has its costs and benefits, as a double-edged sword. In the case that CSR brings higher benefits than costs, the CSR–CFP relationship can be positive. Previous studies have detected the key influencing factors of the CSR–CFP relationship, including firm size [21], corporate governance [22], innovation [23], political embedding [24], and management efficiency [25]. Nevertheless, limited efforts have been made to explore the role of the DOI in the CSR–CFP relationship. Thus, the understanding of the factors behind CR activities conducted by multinational construction firms is incomplete. With this in mind, this study aims to address two fundamental questions as follows:

(1) Does CSR have an inverted U-shaped effect relationship to CFP?

(2) Can DOI moderate the CSR–CFP relationship?

In recent years, Chinese construction companies have accelerated the pace of internationalization and carried out more and more international projects. However, these companies have been widely criticized for their sluggish CSR performance, with serious negative issues, such as environmental pollution and product quality, being revealed [26]. Therefore, their fulfillment of CSR still needs to focus on societal concerns. This study employs Chinese-listed construction firms as a sample to address the above two questions. The results obtained after the analysis show that a higher CSR level does not always have a better CFP level, and the DOI weakens the inverted U-shaped relationship between CSR and CFP. Consequently, the current research promotes the understanding of the impact of CSR and DOI on CFP, and provides guidelines for construction firms to implement CSR in the context of internationalization.

The paper is divided into separate sections based on content. In the next section, the possible CSR–CFP relationship and the importance of considering DOI as a firm-specific factor that shapes it are described. The Methods Section highlights the panel data method for evaluating the inverted U-shaped relationship and moderating effect. In the Results Section, the statistical analysis confirms how different levels of CSR create diverse impacts on financial performance and the moderating influence of DOI levels on the CSR–CFP relationship. The Discussion Section analyzes and develops an understanding of the complex relationship between CSR, DOI, and CFP. Finally, the Conclusion emphasizes the theoretical and methodological contributions of the analytical methods and outlines the study's research limitations and future research directions.

## 2. Theoretical Background

### 2.1. The CSR–CFP Relationship

CSR is an evolving and dynamic concept [27]. Sheldon [28] first proposed the concept of CSR as the contribution of enterprises to meeting the needs of internal and external affiliates. Specifically, Bowen et al. [29] defined CSR systematically and developed the "contribution" as the social goals and values that enterprises should consider when making decisions and taking actions. Moreover, Davis [30] argued that CSR includes both economic and non-economic attributes. On the other hand, others offered a narrow concept of

CSR, which excludes the economic and legal responsibilities [31,32]. There has been no conclusion about the concept of CSR, and since then, some scholars have tried to explain CSR from a more comprehensive perspective [33–35]. According to the European Commission (2011), it incorporates social, environmental, ethical, human rights, and consumer concerns into a firm's business process. The implementation of CSR maximizes firms' shared value for stakeholders and welfare for society, concurring with the stakeholder theory that CSR should extend beyond shareholders to meet different needs, such as the external environment, consumers, employees, and communities [36,37].

Business activities are often motivated by profit seeking. The correlation between CSR and enterprise economic profits has been researched for a long time by McWilliams [38]. Most previous studies indicated that CSR activities impact positively on business profitability [39,40], and firms may promote CFP activities through a proper CSR fulfillment [41]. CSR activities convey positive information to main stakeholders and increase their benefits or power to bring substantial rewards. One of the reasons is that stakeholders have a higher willingness to cooperate with firms with a higher CSR performance [42]. Thus, firms can gain tangible and intangible benefits from engaging in CSR, such as a reduction in employee turnover rate, an increase in employee commitment, customer loyalty, and satisfaction [37], improvements in corporate image and reputation [7], and development of the supply chain and community and government relations [43].

Nevertheless, CSR implementation poses costs to business operation, diversify creditors' energy [44], and consumes corporate resources, such as human capital, cash, and facilities [45]. In this respect, agency theory suggests that managers tend to improve their benefits, such as using philanthropy and donation activities to enhance personal reputation or acquire promotions, instead of taking practical measures to improve stakeholders' benefits [46]. Therefore, an agency relationship may result in high and unnecessary costs. In this case, excessive CSR fulfillment inevitably transfers CSR expenditures to other stakeholders in the form of higher product prices and lower employee salaries [38]. Similarly, the realization of these issues by stakeholders results in further reductions in resource inputs or other stringent controls. Further, the stakeholders of firms conducting greater than required CSR efforts will signal the availability of idle resources [47], which gives stakeholders an unfavorable impression of mismanagement and further reduces the enthusiasm for cooperation between the two parties.

Although most companies have much interest in CSR matter, they may not own sufficient human resources and capital for a better performance [48]. Investors in nature pursue profits, and it is increasingly important for managers to understand the CSR–CFP relationship [49]. The body of evidence accumulated about the nature of the CSR–CFP relationship is equivocal. These conflicting results are mainly due to industry effects [15] and national background [50]. Additionally, CFP quantification is unsuitable for making a cross-industrial comparison. Moreover, compared with developed countries, those emerging market countries have shortcomings in their multinational operation and CSR performance. Consequently, there is uncertainty pertaining to whether the conclusion based on developed countries' backgrounds applies to developing countries [51]. In effect, while most previous studies have tested the linear CSR–CFP relationship, some studies demonstrated that the relationship is non-linear [52]. In this regard, the inverted U-shaped relationship demonstrates that a better CSR performance enhances CFP to a higher extent, but excessive inputs of CSR resources might reduce corporate profits [7,45]. It seems that stakeholders cannot give firms unlimited resource supports [52,53].

### 2.2. Moderating Role of Internationalization

It is of paramount importance to understand the factors influencing the CSR–CFP relationship in a competitive environment [54]. Over time, firms have started to bear tremendous pressure to conduct business in a socially responsible manner due to environmental pollution and resource shortages in international expansion [55]. A multinational business structure helps firms to strengthen CSR commitments [56] and recognize the

contributions of internationalization and implementation of CSR for a satisfactory market position. The complexity of politics, economy, culture, and law in different countries tends to expand the identities of stakeholders in firms' external environment [57]. Consequently, operating in a multinational context requires firms to meet the needs of a wider range of stakeholders [16]. However, engaging all the stakeholders in the process of CSR fulfillment is highly challenging [17]. The challenges are posed by the fact that multinational operations have to cope with novel and unfamiliar responsibilities.

Regarding multinational firms, CSR implementation is assumed to reduce business operation costs and improve the acceptance and legitimacy of stakeholders [58]. Nevertheless, it may unexpectedly result in a distrust of stakeholders [54]. Firms starting business operation in the international market have complex relationships with stakeholders due to little knowledge about laws, regulations, and systems of business in foreign markets [59]. An experienced firm knows to decrease the friction in business activities, principles and culture, and further reaps benefits from fulfilling CSR. In reverse, better financial performance allows multinational firms to continue to operate in overseas markets [60] and maintain competitiveness.

The complicated mechanism of business operation and external transaction costs ascend with DOI. In the case that firms make fewer efforts to implement CSR, then it may cause prejudice and bias among local stakeholders [19]. In this way, the skepticism among stakeholders forces firms to pay higher costs [61]. For example, more intensive searching, extended negotiation, frequent communication, unfavorable contract terms, and litigation can be encountered, leading to a negative impact on financial performance [62]. As an essential part of competition strategies, international business consumes corporate resources and costs. Most firms ought to improve CSR performance, and satisfy the requirement and expectation of greater stakeholders to ensure their standing. They can convey valuable information about firms' overall reliability, stability, and credibility to stakeholders through implementing CSR [63]. The effective cooperation with stakeholders helps firms to gain overseas recognition, attract stakeholders' attention, and bring positive financial influence [58]. Thereby, business operation costs can be under control [64].

## 3. Research Methods

### 3.1. Sample and Data Collection

China has a unique feature regarding government regulation and business system. Its deepening integration into global economies provides a new environment for CSR fulfillment [65]. In effect, the increased firm-scale and domestic market saturation have prompted these organizations to focus on international operations and CSR activities. Thus, Chinese construction firms' probe into international business provides an appropriate opportunity for this study to detect the relationship between CSR, DOI, and CFP. Panel data covering Chinese construction firms listed on the Shanghai Stock Exchange (SHSE) or Shenzhen Stock Exchange (SZSE) from 2010 to 2020 were thus targeted.

The data were collected from three sources: the Wind database, Hexun database, and the sample firms' annual financial reports. We retrieved CSP scores from the Hexun database, an authoritative third-party rating agency in China. The database offers CSR evaluation results based on CSR reports and annual reports of listed firms in China since 2010. It also contains up-to-date information to make the distribution of scores reasonably stable over time. The primary operational and financial data source is the Wind database, which is China's most authoritative financial database. This study included 126 listed construction firms. The exclusion of the missing data and the "ST" label during the sample period [51] resulted in providing a sample of 72 firms and 501 firm annual observations.

### 3.2. Variables

Dependent variable. The return on equity (ROE) was deemed the proxy for the CFP, referring to the percentage rate of net profit divided by net assets. It is an effective measure

of the firm's ability to maximize profit from a capital investment [66] and sample firms' ROE mainly ranges from 0–20%.

Independent variable. CSR represents the firms' responsibility towards society. However, CSR cannot be measured easily due to its abstractness and inclusiveness [26]. Hence, previous studies have proposed corporate social performance (CSP) to represent a measurable form of CSR [67]. Carroll [68] suggested that CSP represents outcomes and results, so it was adopted as a proxy for CSR fulfillment in the study. The Hexun CSR Evaluation Database provides complete CSP information of Chinese firms. It evaluates the CSP of listed firms based on the information obtained from the firm's annual financial reports or non-financial reports. There are five dimensions in the evaluation system, including various indicators and sub-indices. The evaluation indicators of the construction industry are shareholders (30%); employees (15%); suppliers, customers, and consumers (15%); environment (20%); and community (20%). According to statistics data, the CSP scores of Chinese construction firms fall into the interval between 10–30 points. There are approximately 13% of firms with CSP scores between 50–80 and approximately 2% in the range of 30–50.

Moderator variable. We used the percentage of operating income outside mainland China in total income to stand for the DOI [18], where foreign sales are the sum of sales of all foreign market segments [69]. In the case the ratio is close to 1, then the firm becomes highly dependent on the international market. In our study, most of the sample's overseas market turnover firms accounted for less than 10% of the total turnover.

Control variables. We included the operating and financial influencing factors as control variables. First, the size of a firm has a significant impact on both CSR and CFP. Large firms usually have richer resources and funds for CSR investment; they are faced with more significant challenges [70] and external pressure [71]. In addition, firm size may significantly impact CFP [72], which was represented through "total assets" and "the number of employees" [73]. The former explains financial scales and the latter describes organization scale. The logarithmic treatment of a firm's total assets [74] was taken along with "net debt" as another financial variable. Further, firm age was seen as a critical variable.

### 3.3. Model

We adopted regressive modeling to conduct the statistical analysis [75]. Before conducting the regression analysis, the interaction and quadratic terms of continuous variables were confirmed and the correlation was checked to verify that the data were not affected by multicollinearity. Following Mela and Kopalle [76], the correlation coefficient (0.7) was regarded as the threshold for the existence of collinearity. Consequently, no correlation coefficient exceeded 0.7. Further, a VIF test to check whether the sample had multicollinearity problems was conducted. The VIF test showed that none of the variables had a variance inflation factor higher than 5, and thus the research samples were not affected by multicollinearity.

Subsequently, the Breusch and Pagan Lagrange multiplier test and the Hausman test were run to decide whether it was appropriate to use a combined OLS model, a fixed-effects model, or a random-effects regression model [77]. These three models are highly relevant and suitable for studying the data set in which each cross-sectional unit repeats over time and solving the problem of missing variables [78]. The test results showed that fixed-effects panel data analysis was suitable for regression modelling, and the equation is as follows:

$$CFP_{it} = \beta_0 + \beta_1 CSR_{it} + \beta_2 CSR_{it}^2 + \beta_3 DOI_{it} + \beta_4 CSR_{it} * DOI_{it} + \beta_5 CSR_{it}^2 * DOI_{it} + X_{nit}'^{\beta_n} + \varepsilon \tag{1}$$

where $CFP_{it}$ refers to the ROE of firm $i$ at time $t$; $CSR_{it}$ represents the CSP score of the firm $i$ at time $t$; and $CSR_{it}^2$ is the quadratic parameter of $CFP_{it}$. $DOI_{it}$ represents the proportion of the firm's overseas operating income to the primary operating income measured at time $t$. $CSR_{it} * DOI_{it}$ measures the interaction between the CSP score and the DOI, and

$CSR_{it}^2 * DOI_{it}$ measures the interaction between the secondary CSP score and the DOI. The vector $X'_{nit}$ includes the control variables.

## 4. Results

Table 1 shows the descriptive statistics. The average DOI is 14.8% with a minimum of 0% and a maximum of 98.3% limits. In addition, firms' ROE performance with an average ROE of 8.8% ranges from −67.8% to 56.8%. The average lntotal assets is 23.62, while the number of employees ranged between 147 and 552,810, with an average of 43,343. Firm age ranges 2–38 years, with an average age of 17.303 years.

**Table 1.** Descriptive statistics.

| Variable | Obs | Mean | Std. Dev. | Min | Max |
|---|---|---|---|---|---|
| CSR | 501 | 25.515 | 16.501 | −7.17 | 77.92 |
| DOI | 501 | 0.148 | 0.202 | 0 | 0.983 |
| lnTotalAssets | 501 | 23.6160 | 1.8463 | 19.5852 | 28.6365 |
| NetDebt | 501 | 14,570,000,000 | 44,470,000,000 | −22,730,000,000 | 343,600,000,000 |
| Age | 501 | 17.303 | 6.919 | 2 | 38 |
| ROE | 501 | 0.088 | 0.108 | −0.678 | 0.568 |
| Employees | 501 | 43,343.323 | 101,298.53 | 147 | 552,810 |

We employed Pearson's correlation to examine the potential bivariate relationship between the model's variables (Table 2). As shown, ROE has a significant correlation with all variables at the 5% significance level, excluding net debt and employees. Further, CSR is related to all variables at the 5% level of significance.

**Table 2.** Correlations.

| Variables | (1) | (2) | (3) | (4) | (5) | (6) | (7) |
|---|---|---|---|---|---|---|---|
| (1) ROE | 1.000 | | | | | | |
| (2) CSR | 0.381 *** | 1.000 | | | | | |
| | (0.000) | | | | | | |
| (3) lnTotalAssets | 0.095 | 0.264 *** | 1.000 | | | | |
| | (0.033) ** | (0.000) | | | | | |
| (4) NetDebt | 0.015 | 0.101 ** | 0.649 *** | 1.000 | | | |
| | (0.735) | (0.024) | (0.000) | | | | |
| (5) Age | −0.156 *** | −0.311 *** | −0.331 *** | −0.280 *** | 1.000 | | |
| | (0.000) | (0.000) | (0.000) | (0.000) | | | |
| (6) Employees | 0.051 | 0.174 *** | 0.582 *** | 0.487 *** | −0.330 *** | 1.000 | |
| | (0.256) | (0.000) | (0.000) | (0.000) | (0.000) | | |
| (7) DOI | 0.095 * | 0.102 ** | −0.057 | −0.003 | 0.044 | 0.026 | 1.000 |
| | (0.034) | (0.022) | (0.199) | (0.946) | (0.326) | (0.556) | |

Note: *, **, and *** indicate significance at the 0.05, 0.01, and 0.001 levels, respectively.

Five models were detected and the results are given in Table 3. Model 1 regressed the moderating variable and control variables on the CFP. Following, CSR and $CSR^2$ were added to the regression in Model 2 and Model 3. Model 4 tested the moderating effect of DOI on the linear CSR–CFP relationship, while Model 5 tested the moderating effect of DOI on the curve relationship between CSR and CFP.

**Table 3.** Regression results.

| ROE | Model 1 | Model 2 | Model 3 | Model 4 | Model 5 |
|---|---|---|---|---|---|
| CSR | | 0.0018 ***(0) | 0.0062 ***(0) | 0.0017 ***(0) | 0.0062 ***(0) |
| $CSR^2$ | | | −0.0002 ***(0) | | −0.0002 ***(0) |
| Interact1 | | | | 0.0016(0.243) | −0.0032(0.167) |
| Interact2 | | | | | 0.0002 **(0.017) |
| DOI | 0.0490(0.273) | 0.0744 *(0.089) | 0.0511(0.189) | 0.0645(0.147) | 0.0071(0.865) |
| lnTotalAssets | 0.0797 ***(0) | 0.0693 ***(0) | 0.0509 ***(0) | 0.0681 ***(0) | 0.0512 ***(0) |
| NetDebt | 0.0000(0.131) | 0.0000(0.261) | 0.0000(0.184) | 0.0000(0.293) | 0.0000(0.237) |
| Year | −0.0153 ***(0) | −0.0101 ***(0) | −0.0079 ***(0.001) | −0.0101 ***(0) | −0.0080 ***(0.001) |
| Employees | 0.0000(0.63) | 0.0000(0.298) | 0.0000(0.894) | 0.0000(0.357) | 0.0000(0.81) |
| Constant | −1.5391 ***(0) | −1.4419 ***(0) | −1.0969 ***(0) | −1.4090 ***(0) | −1.0925 ***(0) |
| Effect | Fixed | Fixed | Fixed | Fixed | Fixed |
| R-squared | 0.0851 | 0.1386 | 0.3222 | 0.1414 | 0.3340 |
| F-test | 7.846 | 11.293 | 28.516 | 9.884 | 23.293 |
| Prob > F | 0.0000 | 0.0000 | 0.0003 | 0.0000 | 0.0004 |
| Rho | 0.8220 | 0.8052 | 0.6997 | 0.7970 | 0.6830 |

Note: Panel data regression (fixed-effects) with ROE as dependent variable. Z-statistics are in parentheses. *, **, *** denote significance at 10%, 5% and 1% levels, respectively.

Models 3 and 5 were utilized to test the inverted U-curve relationship of CSR–CFP and the moderating effect of the DOI. The results given in Table 3 demonstrate an inverted U-shaped curve relationship between CSR and CFP (Model 3) and provide evidence for the negative regulation of $CSR^2$. Although the coefficient of $CSR^2$ in Model 3 is significant, it is not rigorous enough to demonstrate the inverted U-shaped curve relationship [75]. Therefore, we conducted a U-test to further test the inverted U-shaped curve according to the method of Lind and Mehlum [79]. In this regard, the slope of the lower limit must be negative and significant, while the slope of the upper limit should be positive and significant to satisfy the inverted U-shaped relationship. In addition, the extreme point must be between the two endpoints of the curve (Table 4). The results verify the inverted U-shaped relationship between CSR and CFP.

**Table 4.** Test for the U-shaped curve.

| | Lower Bound | Upper Bound |
|---|---|---|
| Interval | −7.170 | 77.920 |
| Slope | 0.009 *** | −0.019 *** |
| | (17.072) | (−12.812) |
| | t-value | *P*-value |
| Overall test | 12.81 | 0.0000 |
| Extreme point: 19.70382 | | |

Note: t-values are in parentheses. *** denote significance at 1% levels, respectively.

Figure 1 shows that, based on CFP values, firms located on the left side of the curve show an exponential upward trend with a rise in the CSR level. The firms on the left side of the curve exhibit an exponential downward trend with an increasing CSR level. Model 5 outlines the significant positive interaction between DOI and $CSR^2$. The positive interaction suggests that the U-shaped curve becomes smoother [75], as shown in Figure 2.

Lastly, a t-test was performed as a robustness check to verify the results (Table 5). In addition, Heckman procedure [80] (Table 6) was performed to check whether there are any sample selection biases in the overall construction industry. The first stage of the test included a correctional variable outside the final model [81]. Based on the study by Maon et al. [82], the geographic location (province) of firms' headquarters was selected as the exclusion limit in the first-stage equation of the Heckman model, as this factor affects firms' CSR fulfillment manner. Moreover, the average ROE of the sample is not different from the population of 649 firms, and the sample is fully representative of the population. Table 6

shows the results of the two stages of Heckman. The lambda coefficients of Models 3 and 5 are significant, which indicate that the results may be affected by sample selection bias. However, the revised regression results are consistent with the original ones.

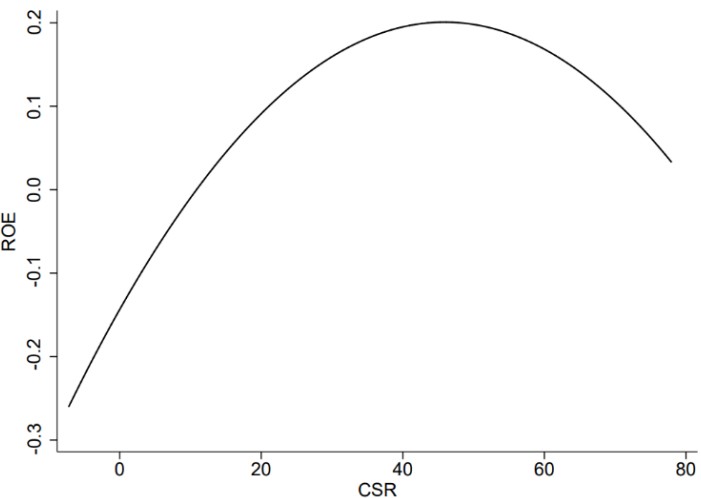

**Figure 1.** The U-shaped relationship between CSR and CFP.

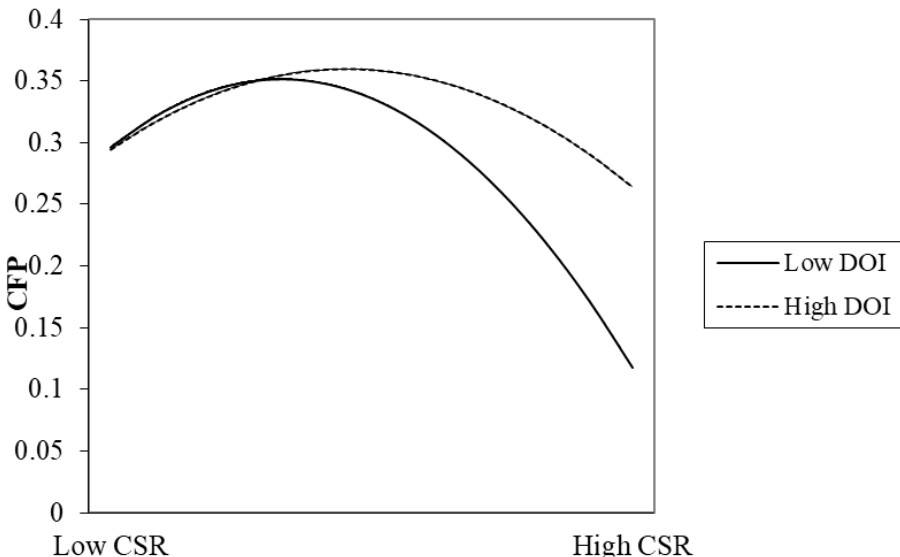

**Figure 2.** The U-shaped relationship between CSR and CFP, moderated by DOI.

**Table 5.** Robustness test 1.

| Group | Observations | Mean | Std. Dev. |
|---|---|---|---|
| obs1 | 501 | 0.0959 | 0.0958 |
| obs2 | 498 | 0.0884 | 0.1083 |
| Combined | 999 | 0.0921 | 0.1023 |
| Ha: diff < 0 | Ha: diff! = 0 | Ha: diff > 0 | |
| Pr(T < t) = 0.8766 | Pr(|T| > |t|) = 0.2469 | Pr(T > t) = 0.1234 | |

Note: *t*-test between censored and uncensored observations for ROE (dependent variable of our regressions).

**Table 6.** Robustness test 2.

| ROE | Model (3) | Model (5) |
|---|---|---|
| CSR | 0.0069 ***(0) | 0.0070 ***(0) |
| $CSR^2$ | −0.0002 ***(0) | −0.0002 ***(0) |
| Interact1 | | −0.0028(0.193) |
| Interact2 | | 0.0002 ***(0.009) |
| DOI | 0.0334 *(0.079) | −0.0032(0.887) |
| lnTotalAssets | 0.0010(0.734) | 0.0016(0.593) |
| NetDebt | 0.0012(0.152) | 0.0000(0.132) |
| Year | −0.0012 *(0.051) | −0.0012 **(0.043) |
| Employees | 0.0000(0.411) | 0.0000(0.437) |
| Wald chi$^2$ | 339.970 | 356.390 |
| Prob > chi$^2$ | 0.0003 | 0.0000 |
| Rho | 0.27743 | 0.27639 |
| lambda | 0.0237 *(0.057) | 0.0233*(0.059) |

Note: OLS, with Heckman procedure, with the independent and the moderator variables and their interactions. ROE is the dependent variable. t-statistics are in parentheses. *, **, *** denote significance at 10%, 5% and 1% level, respectively.

Considering the influence of the *DOI*, the inflection point of the inverted U-shaped curve was quantified and the movement of the inflection point can be outlined. The inflection point corresponds to the axis of symmetry of the quadratic function. Equation (1) represents the inflection point ($CSR^*$). The partial derivative of $CSR^*$ to *DOI* was obtained (Equation (3)) to test the influence of the *DOI* on the inflection point $CSR^*$. If the partial derivative is greater than 0, the inverted U-shaped curve moves to the right as the DOI increases. The denominator in Equation (3) is greater than 0. The sign of the formula depends on the numerator. Substituting the regression result into Equation (3), $CSR^* > 0$ is derived. The inflection point $CSR^*$ of the inverted U-shaped curve shifts to the right as the *DOI* increases. According to the different values under high and low *DOI*, the maximum value of CFP at the inflection point of the inverted U curve is obtained. The maximum CFP value under high *DOI* is significantly higher than that under low *DOI*.

$$CSR^* = -\frac{\beta_1 + \beta_4 DOI}{2(\beta_2 + \beta_5 DOI)} \tag{2}$$

$$\frac{\partial CSR^*}{\partial DOI} = -\frac{\beta_1 \beta_5 - \beta_2 \beta_4}{2(\beta_2 + \beta_5 DOI)^2} \tag{3}$$

## 5. Findings and Discussion

### 5.1. The CSR–CFP Relationship

The CSR–CFP relationship presents an inverted U-shaped curve. The results are similar to those in the studies of [10,11], and contrary to the U-shaped relationship between CSR–CFP as verified by [8,9]. The research findings cannot support a linear relationship between CSR and CFP as highlighted in previous studies. Model 5 indicates that the CSR variable's quadratic coefficient is significantly negative. This result concurs with previous literature results [83]. The inverted U-shaped curve shows that the CSR–CFP relationship develops from positive to negative. Supposing that CSR efforts are low, CSR and CFP should be positively correlated, and CFP is in the middle position. Firms with moderate CSR efforts may obtain a higher CFP level, which enables managers to obtain an ideal CSR engagement scope. However, for firms that strive to reach a higher CSR level, excessive CSR activities led to profits being unable to compensate for CSR investment. The marginal cost of CSR in the initial stage is greater than the marginal benefit; thus, CSR can improve CFP in the initial phase of performance. In this case, the marginal cost equals the marginal revenue, and the CFP reaches the threshold. However, when CSR efforts exceed the threshold, a downward trend in the CFP can be identified. Unlike other industries, construction projects have a one-off feature related to firm stakeholders. The

stakeholders of construction enterprises include government, subcontractors, suppliers, and communities. The key factors or activity areas of CSR are environmental protection, construction quality and safety, community, employees, clients, and CSR management [84].

Wnuczak [42] and Orlitzky and Shen [85] stated that stakeholders are willing to cooperate with companies with higher CSR performance. However, the research findings cannot establish the preferences of the stakeholders to reward firms having higher levels of CSR scores despite making substantial CSR efforts and costs. This can be explained through the use of stakeholder and agency theories. In this respect, it is considered that firms establish associations and cooperation with stakeholders by means of CSR implementation. Stakeholders tend to develop relationships with those firms that have a good CSR performance [86], which will in turn help construction firms to improve their financial performance. The limited resources disallow stakeholders to provide permanent support to firms, and the positive response reduces as the CSR efforts reach a certain level. Once the cost exceeds its positive return, it has a negative impact on financial performance. In this regard, the agency theory highlights that managers tend to improve their own benefits and ignore or even harm the benefits of other internal and external stakeholders. The excessive CSR investment causes stakeholders to doubt the corporate management resulting in a reduction in the input or support resources. Therefore, CSR efforts below or above a specific limit are undesirable for the sustainable development of construction firms. This finding corrects the misconception of the construction industry that the continuous efforts of construction firms in terms of CSR are not necessarily the most beneficial [87].

*5.2. The Moderating Effect of DOI*

As given in Table 6, Model 5 shows that DOI weakens the inverted U-shaped relationship between CSR and CFP. The findings differ from the negative synergies of DOI and CSR as reported in the literature [15], showing more complex moderating effects. However, in an international context, there is no guarantee that CSR deserves investment [88]. The moderating effect of DOI reveals that CSR and CFP's non-linear relationship in low DOI firms is more significant while the curve is steeper. In the case of a low CSR, construction firms with a low DOI under the same CSR effort level have a higher CFP. Once CSR exceeds the threshold, construction firms with a high level of internationalization under the same CSR effort level have a higher CFP. In addition, firms with a higher DOI need to pay more CSR efforts to achieve a higher level of profitability.

While both business and academic circles highlight the importance of multinational operations and CSR activities, there is a lack of research combining these two factors. Multinational firms have to face trade barriers [89] and they find it challenging to establish good cooperative relations with local stakeholders, which results in more transaction and agency costs [90]. In this study, an analysis of the impact of DOI on the CSR–CFP relationship was conducted. Firms face more complex stakeholder relationships in overseas markets [57], and fulfilling social responsibilities can release positive corporate information to stakeholders. In the case of construction firms starting in the international market, fulfilling CSR activities can help to gain stakeholder support and benefits. However, with a higher DOI, low-level CSR activities cause challenges in meeting the needs of external stakeholders and establishing a favorable reputation in overseas markets for preferential policies. Accordingly, it is necessary for firms to provide greater CSR efforts to reap the benefits based on legitimacy continuously.

A firm with a high DOI fulfils more CSR efforts and breaks through the original trade barriers, as it has the capability to fully integrate and utilize the knowledge and resources of stakeholders in different markets [91]. Consequently, these firms have more potential for profits. Once a firm breaks the threshold, costs exceed revenue with increasing CSR levels, so CFP shows a downward trend. Firms with a higher DOI have fewer adverse effects caused by excessive CSR investment. Moreover, firms' CSR activities help to reduce frictions between host countries and multinational firms through CSR. They can be more standing in overseas markets and establish corporate business networks [92] to enhance

their capabilities and generate a higher wealth through CSR fulfilment. This weakens the adverse effects of excessive CSR investment, explaining the weakening effect of the DOI on the CSR–CFP relationship in the later period.

*5.3. Managerial Implications*

The inverted U-curve relationship is vital for construction firms to improve practices. The derivation of such a curve indicates that a simply positive or negative linear relationship as found in previous studies [93,94] might not be an effective guidance to corporate management. In this paper, the implication is that blind investments in CSR efforts or lack of suitable efforts can cause disadvantages for development. This study showed that greater CSR efforts do not necessarily improve the CFP, as excessive CSR efforts may be detrimental to CFP. Therefore, there exist some drawbacks of CSR investment without taking account of all stakeholders' needs. Managers need to understand the extent to which the investment is most beneficial for firms. McWilliams [38] found it feasible for firms to optimize CSR efforts by weighing benefits and costs. Further, Lin et al. [95] indicated that firms should provide a certain level of CSR to maximize profits. Therefore, managers are recommended to find a suitable CSR interval and strike the balance between CSR and CFP to utilize corporate resources.

Considering the complexity of the relationship, firms need to consider their specific competence in the CSR implementation process. These factors may determine the effect of each firm's CSR performance on CFP. This study shows that firms with a higher DOI should concentrate more on the impact of CSR and gain more benefits available. Stakeholder management is considered an essential factor in project management [96] and the stakeholder theory explains the importance of implementing CSR to meet stakeholders' requirements for the firm's broader social actions [97]. Engagement in CSR activities enables multinational firms to strengthen their relationships with stakeholders [98]. Thus, corporate managers should pay attention to stakeholder-oriented CSR efforts and give importance to the balance between internal and external stakeholders [99].

The moderating effect of the DOI on the CSR–CFP relationship shows that international construction companies shall understand the CSR–CFP relationship caused by multinational operations, and they need to implement CSR carefully to improve resource utilization efficiency. Regarding the uncertainty of business internationalization, the research findings help construction companies to recognize better the interaction between the DOI and the CSR–CFP relationship. This study contributes to overseas CSR fulfillment by embracing the complexity of CSR implementation in international construction companies.

## 6. Conclusions

CSR is essential for firms to gain legitimacy. Regarding multinational firms, the interference of various factors makes the CSR–CFP relationship complicated, and CSR plays a crucial role in determining the CFP. In this study, an inverted U-shaped curve relationship between CSR and CFP is constructed instead of a simple linear relationship. There may be a scale effect on construction firms' CSR performance. For example, in the case of low-level CSR, the CFP grows with the increase in CSR level while for the high-level CSR, there can be a decline in the CFP. Further, the DOI has a significant moderating effect on the CSR–CFP relationship. Thus, firms with a higher DOI may have tremendous profit potentials.

The research has two main contributions. First, it combined agency and stakeholder theories to consider the CSR–CFP relationship. Under a one-sided understanding, the impact of CSR on CFP is subject to incompletion. Combining the two theories, this study highlighted that the CSR–CFP relationship has a complex non-linear relationship. This inverted U-shaped relationship implies that firms must consider both costs and benefits while fulfilling social responsibilities. Second, the study considered the impacts of DOI on the CSR–CFP relationship from the perspective of stakeholders. The high DOI of multinational firms has significantly weakened the effect of CSR on CFP and caused them

to face more stakeholders. The relationship between the firm and stakeholders seriously affects the CSR–CFP relationship. This finding reveals that the combinations of a high DOI-high CSR and a low DOI-low CSR are more conducive to promoting the CFP of construction firms. In this perspective, necessary enlightenment is that the discussion of the position of the inflection point of the inverted U-shaped curve helps to gain a deep insight into how the DOI of a firm matches CSR and promotes the improvement of corporate performance.

The research has a few limitations. The performance of social responsibilities is examined among Chinese construction firms, and the conclusions may not necessarily apply to firms in other regions or countries. It is expected that future research may consider the influence under a global context.

**Author Contributions:** Conceptualization, M.S. and W.J.; methodology, M.S. and Y.Z.; software, M.S. and Y.Z.; validation, W.J., Y.Z. and K.Y.; formal analysis, M.S.; investigation, M.S. and Y.Z.; resources, M.S.; data curation, M.S.; writing—original draft preparation, M.S.; writing—review and editing, W.J. and K.Y.; visualization, M.S.; supervision, W.J.; project administration, W.J. and K.Y.; funding acquisition, W.J. All authors have read and agreed to the published version of the manuscript.

**Funding:** This research was funded by the Humanities and Social Science Foundation of the Ministry of Education of China (Grant Number 19YJC630065).

**Data Availability Statement:** Data are available from the authors upon request.

**Conflicts of Interest:** The authors declare no conflict of interest.

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
