# Peer review of "Moderating Effects of Internationalization between Corporate Social Responsibility and Financial Performance: The Case of Construction Firms"

_buildings, doi:10.3390/buildings12020185_

Round 1

Reviewer 1 Report

I propose some minor improvements to the exposition and discussion of the results. 

  1. Row number 11 onwards to the end of the abstract: Please write the section of the abstract from the sixth sentence, row number 11 in present tense. For example write "In this study, an analysis of the impact of CSR on the financial performance of construction firms is conducted and..." Please do not use past tense in the remainder of the abstract like has been conducted. It just does not sound right.
  2. in row number 55 Please delete one unnecessary sentence: "However, once the costs exceed the benefits it becomes harmful to invest in CSR activities." This does not add any information to the text.
  3. in row 64 replace "It employs" with "This study" employs
  4. In the conclusions starting in row 410. You summarise the results in the conclusions, but please add a sentence about "Why it is interesting to specifically observe a nonlinear relationship between CSR and CFP for construction firms! In the introduction you mention that construction firm has a great impact on the environment and that they operate in multinational, global markets. Hence say something about what construction firms are particularly interesting in this context. If you wish this statement could also be at the end of the introduction (before the structure of the paper). Please do not hesitate to ask if any of these instructions are unclear.

Author Response

Dear reviewer, we have revised the paper according to your suggestions:
1. We have changed the specific sentences to present tense.
2. We have deleted this unnecessary sentence.
3. We have replaced "It employs" with "This study" employs.
4. We have added the discussion about what construction firms are particularly interesting in this context at the end of the introduction (before the structure of the paper).

Reviewer 2 Report

Thank you for possibility to read this interesting paper. I evaluate it as quite professional written according to the academic rules. However some improvements are necessary:

There is lack of clear aim and the methodology description in the abstract.

The literature review can be developed and the Authors did not present the hypothesis despite of the chapter title “2. Theoretical background and hypotheses”. Consider some more sources:

Aastha, B., Shazi, S.J. Corporate social responsibility practices in small and medium enterprises (2019) Polish Journal of Management Studies, 19 (1), pp. 9-20.

Kot, S., Brzezinski, S. Market orientation factors in sustainable development and corporate social responsibility. (2015) Asian Journal of Applied Sciences, 8 (2), pp. 101-112.

Kot, S. Knowledge and understanding of corporate social responsibility. (2014) Journal of Advanced Research in Law and Economics, 5 (2), pp. 109-119.

Gaol, F.A.L., Harjanto, K. Impact of selected factors towards corporate social responsibility (CSR) disclosure: Evidence from Indonesia. (2019) Polish Journal of Management Studies, 20 (1), pp. 181-191.

Extend results discussion in relation to previous studies.

More practical recommendations should be presented.

Author Response

Dear reviewer, we have revised the paper according to your suggestions:

1. We have added clear aim and the methodology description in the abstract.

2. We have improved the literature review, especially the part related to the CSR concept.

3. We have extend results discussion in relation to previous studies.

4. We have added a whole paragraph to the practical recommendations section.